# HAIR: Hypernetworks-based All-in-One Image Restoration

## Abstract

Image restoration aims to recover a high-quality clean image from its degraded version. Recent progress in image restoration has demonstrated the effectiveness of All-in-One image restoration models in addressing various unknown degradations simultaneously. However, these existing methods typically utilize the same parameters to tackle images with different types of degradation, forcing the model to balance the performance between different tasks and limiting its performance on each task. To alleviate this issue, we propose HAIR, a **H**ypernetworks-based **A**ll-in-One **I**mage **R**estoration plug-and-play method that generates parameters based on the input image and thus makes the model to adapt to specific degradation dynamically. Specifically, HAIR consists of two main components, i.e., Classifier and Hyper Selecting Net (HSN). The Classifier is a simple image classification network used to generate a Global Information Vector (GIV) that contains the degradation information of the input image, and the HSN is a simple fully-connected neural network that receives the GIV and outputs parameters for the corresponding modules. Extensive experiments demonstrate that HAIR can significantly improve the performance of existing image restoration models in a plug-and-play manner, both in single-task and All-in-One settings. Notably, our proposed model Res-HAIR, which integrates HAIR into the well-known Restormer, can obtain superior or comparable performance compared with current state-of-the-art methods. Moreover, we theoretically demonstrate that to achieve a given small enough error, our proposed HAIR requires fewer parameters in contrast to mainstream embedding-based All-in-One methods. Code is available in supplementary materials.

## 1 Introduction

Image restoration is an important task in the field of computer vision, aiming to reconstructing high-quality images from their degraded states. The presence of adverse conditions such as noise, haze, or rain can severely diminish the effectiveness of images for a variety of applications, such as autonomous navigation (Valanarasu et al., 2022; Chen Yu-Wei, 2023), augmented reality (Girbacia et al., 2013; Saggio et al., 2011; Dang et al., 2020). Therefore, developing robust image restoration methods is of great importance. The use of deep learning in this domain has made remarkable progress, as evidenced by a suite of recent methodologies (Zhang et al., 2017b; Liang et al., 2021; Zamir et al., 2022; Chen et al., 2022b; Li et al., 2023; Dudhane et al., 2024). Nonetheless, the predominant approach in current research is to employ task-specific models, each tailored to address a particular type of degradation (Zhang et al., 2018b; Liang et al., 2021; Zamir et al., 2022; Chen et al., 2022b;a). This tailored approach, while precise, presents a constraint in terms of universality, as it restricts the applicability of models to scenarios with varied or unknown degradations (Zamir et al., 2020a;b; Purohit et al., 2021). To overcome this limitation, many researchers have focused on developing All-in-One image restoration models recently. These models are designed to tackle various degradations using one single model. Pioneering efforts in this area (Li et al., 2022; Zamir et al., 2021; Valanarasu et al., 2022; Potlapalli et al., 2024; Zhang et al., 2023; Dudhane et al., 2024; Conde et al., 2024) have utilized a variety of advanced techniques such as contrastive learning (Li et al., 2022), meta-learning (Zhang et al., 2023), visual prompting methods (Potlapalli et al., 2024; Wang et al., 2023; Dudhane et al., 2024; Li et al., 2023; Conde et al., 2024). These approaches have undoubtedly made substantial contributions to this field.

However, these All-in-One models share a common drawback, i.e. they rely on a single model with fixed parameters to address various degradations. This one-size-fits-all method can hinder

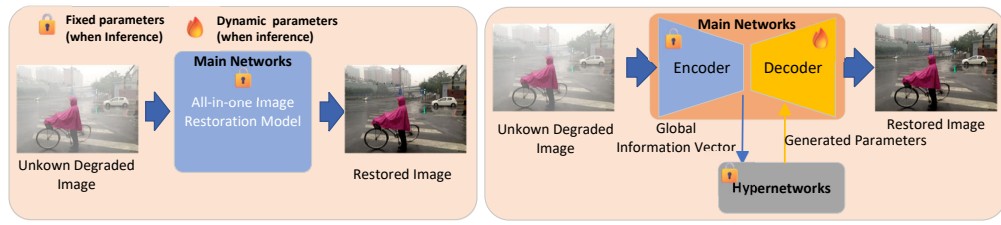

Figure 1: Comparisions between our method and previous methods in the inference stage. (a) Previous All-in-One image restoration methods. These methods utilize a single model with the same parameters to tackle various degradations. (b) Our proposed HAIR. Given a certain degraded image, we first use a fixed encoder to extract the image feature, which is then fed into the Hypernetworks to generate the dynamic parameters for the decoder, and finally obtain the restored image. Note that "dynamic" and "fixed" in this paper are specially for the main networks, as discussed in Appendix A.3.1.

the model's effectiveness when dealing with multiple degradations simultaneously. For example, when viewed through the lens of frequency domain analysis, haze is characterized as low-frequency noise, in contrast to rain, which is considered high-frequency interference. An effective dehazing model acts as a low-pass filter, preserving high-frequency details, whereas deraining requires the opposite—enhancing the high-frequency components. Consequently, a model must balance these conflicting demands of different degradations, thus limiting its performance on each task. We provide more detailed clarifications of this problem in Appendix A.2.1.

To mitigate the aforementioned issue, we propose a **H**ypernetworks-based **A**ll-in-One **I**mage **R**estoration method (HAIR) in this paper. The core idea of HAIR is to generate the weight parameters based on the input image, and thus can dynamically adapt to different degradation information. HAIR employs Hypernetworks (Ha et al., 2016), a trainable neural network, to take the degradation information from the input image and produce the corresponding parameters. Specifically, for a given unknown degraded image, we first utilize a classifier, similar to those used in image classification networks, to obtain its Global Information Vector, which contains crucial discriminative information about various types of image degradation (as shown in Fig. 2 (c-d)). This vector is then used to generate the necessary parameters, as illustrated in Fig. 1. With these dynamically parameterised modules, we ultimately achieve the restored image. In brief, our contributions include:

- We propose HAIR, a novel Hypernetworks-based All-in-One image restoration method that is capable of dynamically generating parameters based on the degradation information of input image. HAIR consists of two components, i.e. Classifier and Hyper Selecting Nets, both of which function as a plug-in-and-play module. Extensive experiments demonstrate that HAIR can significantly improve the performance of existing image restoration models in a plug-and-play manner, both in single-task and All-in-One settings.

- By incorporating HAIR into Restormer (Zamir et al., 2022), we propose a new All-in-One model, i.e. Res-HAIR. To the best of our knowledge, our method is the first to apply data-adaptive Hypernetworks to All-in-One image restoration models. Extensive experiments validate that the proposed Res-HAIR can achieve superior or comparable performance compared with current state-of-the-art methods across a variety of image restoration tasks.

- We theoretically prove that, for a given small enough error threshold $\epsilon$ in image restoration tasks, HAIR requires fewer parameters compared to mainstream embedding-based All-in-One methods like (Li et al., 2022; Potlapalli et al., 2024; Conde et al., 2024).

## 2 RELATED WORKS

### 2.1 ALL-IN-ONE IMAGE RESTORATION

While single degradation methods do achieve great success (Liang et al., 2021; Zamir et al., 2022; Chen et al., 2022b), All-in-One image restoration, which aims to utilize a single deep restoration model to tackle multiple types of degradation simultaneously without prior information about the degradation of the input image, has gained more attention recently (Jiang et al., 2023; Potlapalli et al., 2024; Zhang et al., 2023; Conde et al., 2024; Zamfir et al., 2024). The pioneer work AirNet (Li

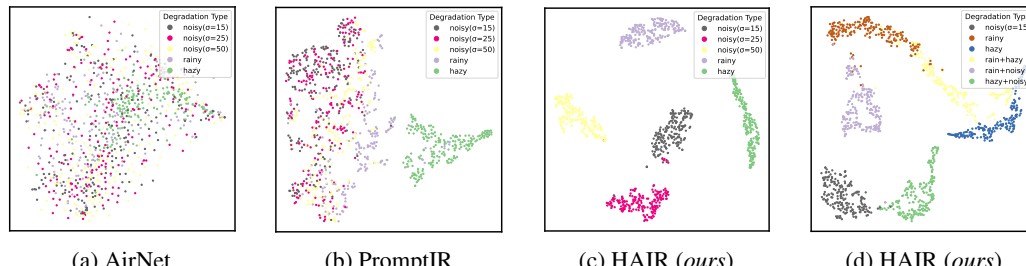

(a) AirNet   (b) PromptIR   (c) HAIR (*ours*)   (d) HAIR (*ours*)

Figure 2: Comparison of tSNE plots for the degradation embeddings between previous methods and our HAIR (i.e. the GIVs). Each distinct color represents a unique degradation type. As shown in (c), our HAIR excels not only in recognizing various degradation types, such as noise, rain, and haze, but also in distinguishing between the same type of degradation at varying intensities, e.g. noise with different standard deviations. Even when confronted with composite degradations not encountered during training, HAIR can also accurately discriminate them, i.e. the GIVs for these composite cases located midway between the GIVs of their constituent degradations, as illustrated in (d).

et al., 2022) achieves All-in-One image restoration using contrastive learning to extract degradation representations from corrupted images. IDR (Zhang et al., 2023) decomposes image degradations into their underlying physical principles, achieving All-in-One image restoration through a two-stage process based on meta-learning. With the rise of LLM (Zhao et al., 2023), prompt-based learning has also emerged as a promising direction in image restoration tasks (Potlapalli et al., 2024; Wang et al., 2023; Li et al., 2023). They typically produce a prompt embedding for each input image based on their content, then inject this prompt into the model to restore the image, which is essentially a conditional embedding. Among them, some works like DA-CLIP (Luo et al., 2023) and InstructIR (Conde et al., 2024) insightfully leverage pre-trained large-scale text-vision models to produce the visual prompts. Next, some methods like DaAIR (Zamfir et al., 2024) and AMIR (Yang et al., 2024) utilize routing techniques, which leverage multiple experts within the model to handle different degradation types or tasks by directing the input data along specialized pathways, to achieve adaptive image restoration. However, the output of these multiple experts is also essentially a kind of conditional embedding. Despite the success these methods have achieved, most of them typically use the same parameters for distinct degradations and only inject the degradation information as conditional embeddings into the model, which forces the model to balance different degradations, and thus impair the performance of the models. Tian et al. (2024) recently employed low-rank matrix decomposition for weight modulation, but their method relies on pre-trained, task-specific weights and prior degradation information, lacking the dynamic parameter generation capability and adaptibility. In contrast, our proposed HAIR can dynamically generate specific parameters for the given degraded image using a hypernetwork and thus making the model adapt to unknown degradations better.

## 2.2 DATA-CONDITIONED HYPERNETWORKS

Hypernetworks (Ha et al., 2016) are a class of neural networks designed to generate weights (parameters) for other networks. They can be classified into three types, i.e task-conditioned, data-conditioned, and noise-conditioned hypernetworks (Chauhan et al., 2023). Among these methods, data-conditioned hypernetworks are particularly noteworthy for their ability to generate weights contingent upon the distinctive features of the input data. This capability allows the network to dynamically adapt its behaviour to specific input patterns or characteristics, fostering flexibility and adaptability within the model. Consequently, this results in enhanced generalization and robustness. Data-conditioned hypernetworks have been applied in many computer vision tasks, e.g., semantic segmentation (Nirkin et al., 2021) and image editing (Alaluf et al., 2022). Despite previous attempts to integrate hypernetworks into image restoration (Aharon & Ben-Artzi, 2023; Fan et al., 2019), these have primarily leveraged task-conditioned hypernetworks, which take a task embedding (like an embedding of derain) as the input and output the weights. They have two main drawbacks: (1) They require prior knowledge of the input image's degradation type. (2) They cannot dynamically generate weights based on image content. Although Klocek et al. (2019) tries to use Data-conditional Hypernetworks in Super-Resolution, it direcly maps a coordinate to a pixel, which is unreliable. To the best of our knowledge, our work is the first to introduce data-conditioned hypernetworks into the domain of All-in-One image restoration.

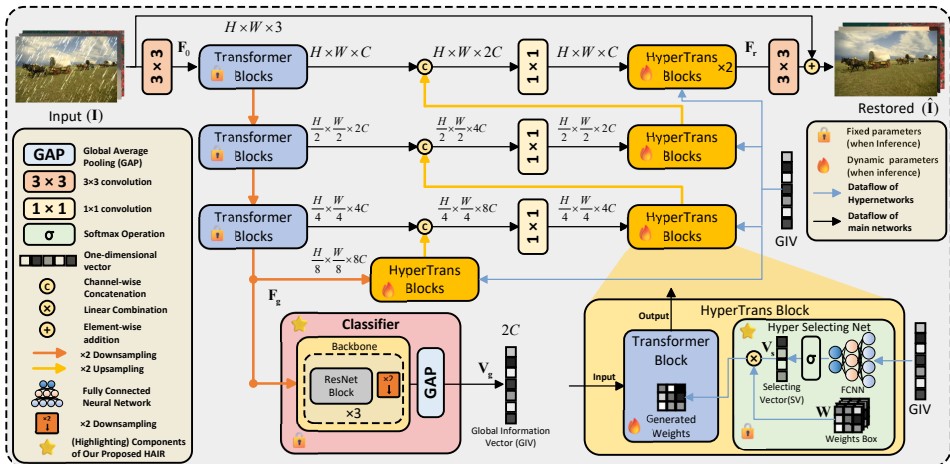

Figure 3: The overall framework of our proposed Res-HAIR. Res-HAIR is built by integrating our HAIR into the popular Restormer (Zamir et al., 2022). HAIR contains two modules, i.e. Classifier and Hyper Selecting Net (HSN). The classifier is used to yield a Global Information Vector (GIV) $\mathbf{V_g}$ from the high-level feature $\mathbf{F_g}$ containing the degradation information of the input image. HSN is used to dynamically generate weights for the Transformer Blocks based on GIV, customizing the restoration process to address the specific degradation characteristics of each image.

## 3 METHOD

In this section, we first introduce our proposed hypernetwork-based All-in-One image restoration module (HAIR) in detail. Then, we propose our All-in-One method (Res-HAIR) by integrating HAIR into the popular Restormer (Zamir et al., 2022). Finally, we provide a brief theoretical analysis on the proposed HAIR module.

### 3.1 HYPERNETWORK-BASED ALL-IN-ONE IMAGE RESTORATION MODULE (HAIR)

In this subsection, we introduce HAIR, a hypernetworks-based All-in-One image restoration module designed to dynamically generate parameters for an image-to-image network based on the input image's features in a plug-and-play manner. HAIR comprises two main components: the Classifier and the Hyper Selecting Net (HSN). Next, we will introduce the two components in detail.

#### 3.1.1 CLASSIFIER

As outlined in Section 1, our approach to extracting degradation information involves designing a straightforward Classifier akin to those used in image classification tasks. Taking Fig. 3 as an example, we start by entering the feature $\mathbf{F_g} \in \mathbb{R}^{\frac{H}{8} \times \frac{W}{8} \times 8C}$ into the backbone, which contains multiple ResNet blocks (He et al., 2016) followed by downsampling $\times 2$. This process is aimed at downsizing the spatial resolution while distilling the essential information, shown as follows:

$$\mathbf{F_g^1} = \text{Downsampling}(\text{ResNetBlock}(\mathbf{F_g})), \mathbf{F_g^t} = \text{Downsampling}(\text{ResNetBlock}(\mathbf{F_g^{t-1}}))), \quad (1)$$

where $t = 1, 2, 3$. After three iterations, we obtain $\mathbf{F_g^3} \in \mathbb{R}^{\frac{H}{64} \times \frac{W}{64} \times 2C}$, which serves as the input of Classifier for generating the global information vector (GIV) $\mathbf{V_g} \in \mathbb{R}^{2C}$ that captures the degraded information. Specifically, GIV is computed as

$$\mathbf{V_g} = \text{GAP}(\mathbf{F_g^3}), \quad (2)$$

where GAP denotes Global Average Pooling. The inclusion of GAP ensures that $\mathbf{V_g}$ remains a one-dimensional vector of consistent size, irrespective of the input image's resolution. Unlike traditional image classification networks, there is no requirement for a Softmax layer here, as its application would confine the values of $\mathbf{V_g}$ within the range [0,1], potentially restricting the parameter generation process. Moreover, the backbone can be substituted with other image classification networks, e.g., VGG (Simonyan & Zisserman, 2014) and Inception (Szegedy et al., 2016), with little negative impact on performance since degradation-type classification (discrimination) is a relatively simple task.

### 3.1.2 HYPER SELECTING NET

After obtaining GIV $\mathbf{V_g}$ from the Classifier, it is then fed into our data-conditioned hypernetworks, i.e. Hyper Selecting Net (HSN), to generate weights for the corresponding modules. Let's still take Fig. 3 as an example. Given $\mathbf{V_g} \in \mathbb{R}^{2C}$, a Selecting Vector $\mathbf{V_s} \in \mathbb{R}^N$ is initially computed as:

$$\mathbf{V_s} = \sigma(\mathbf{FCNN}(\mathbf{V_g})), \tag{3}$$

where $\sigma$ is a Softmax operation and $\mathbf{FCNN}$ denotes a simple fully-connected neural network. Unlike $\mathbf{V_g}$, $\mathbf{V_s}$ will be used to directly generate the parameters, so we need a $\sigma$ to make sure $\mathbf{V_s}$ is positive and $\in [0, 1]$, thus making parameter generation more stable. Subsequently, the parameters $\mathbf{w}$ are derived as

$$\mathbf{w} = \sum_{i=1}^{N} \mathbf{V_s}^i \mathbf{W}_i. \tag{4}$$

In this formula, $\mathbf{V_s}^i$ represents the $i$-th element of $\mathbf{V_s}$. The matrix $\mathbf{W} \in \mathbb{R}^{N \times P}$, referred to as the Weight Box, comprises $\mathbf{W}_i \in \mathbb{R}^P$ as its $i$-th row. The hyperparameter $N$ influences the total number of parameters, with $P$ being the count of parameters required for one corresponding module (i.e. one Transformer Block in this example). With $\mathbf{w}$ determined, it is then used as the parameters of the corresponding module. So the operation of the "HyperTrans Block" in Fig. 3 can be represented as:

$$\mathbf{x}' = \mathbf{Transformer\_Block}(\mathbf{x}; \mathbf{w}). \tag{5}$$

Here, $\mathbf{x}$ is the input and $\mathbf{x}'$ is the output. As the Transformer Block (see Appendix A.4) is based on convolution, $\mathbf{w} \in \mathbb{R}^P$ is reshaped into four-dimensional tensors to serve as convolution kernels for the Transformer Block. To reduce the number of parameters, the transformer blocks at the same decoder level share one weight box, and each transformer block is independently equipped with its own FCNN.

### 3.2 PROPOSED ALL-IN-ONE IMAGE RESTORATION MODEL (RES-HAIR)

Integrating HAIR into an existing image-to-image network involves a simple two-step process: (1) Insert the Classifier at the junction between the first and second halves of the network to produce a Global Information Vector (GIV) from the features of the first half. (2) Incorporate HSNs into the second half of the network, using the GIV to generate weights for all modules within this section dynamically. In this work, we integrate our proposed HAIR module into the popular image restoration model Restormer (Zamir et al., 2022), and propose our All-in-One model Res-HAIR. Although we only use the example of integrating HAIR with Restormer to illustrate the process, this example is representative, and integration with other networks follows in a virtually identical way.

**Overall Architecture.** As shown in Fig. 3, our network architecture is consistent with Restormer (Zamir et al., 2022). Given a certain degraded image input $\mathbf{X} \in \mathbb{R}^{H \times W \times 3}$, Res-HAIR utilizes a $3 \times 3$ convolution to transform $\mathbf{X}$ into feature embeddings $\mathbf{F_0} \in \mathbb{R}^{H \times W \times C}$, where $C$ denotes the number of channels. These feature embeddings are then proceeded through a 4-level hierarchical encoder-decoder, resulting in deep features $\mathbf{F_r}$. Each level of the encoder-decoder incorporates several Transformer blocks, with an increasing number from the top to the bottom level, ensuring computational efficiency. The left three blue "Transformer Blocks" function as an Encoder, designed to extract features from $\mathbf{F_0}$ and ultimately produce a global feature map $\mathbf{F_g} \in \mathbb{R}^{\frac{W}{8} \times \frac{H}{8} \times 8C}$ with a large receptive field. $\mathbf{F_g}$ is then input into a Classifier to yield a Global Information Vector (GIV) $\mathbf{V_g} \in \mathbb{R}^{2C}$. The right four orange "HyperTrans Blocks" operate as a Decoder, aiming to adaptively fuse features at each Decoder level based on degradation, culminating in a restored image $\hat{\mathbf{X}}$. Specifically, at each Decoder level, the HyperTrans Block receives two inputs, i.e. the input feature and the GIV $\mathbf{V_g}$. $\mathbf{V_g}$ is fed into the Hyper Selecting Net to generate weights for the corresponding Transformer Blocks, which are then applied to the input feature to produce the output. It is important to note that all HyperTrans Blocks use the same $\mathbf{V_g}$ derived from $\mathbf{F_g}$. Since the weights in the Decoder are generated based on $\mathbf{V_g}$, which contains the degradation information of $\mathbf{X}$, our method can thus adapt to different degraded images.

### 3.3 THEORETICAL ANALYSIS

In this subsection, we first interpret the generated weights from a selection perspective and then provide a brief theoretical analysis of model complexity.

### 3.3.1 RETHINKING WEIGHT GENERATION VIA A SELECTING PERSPECTIVE

Considering the Weight Box as part of the FCNN, it may not be immediately clear why such a direct generation of weight from a network works well. This seeming lack of intuitiveness is what we aim to address. Before delving into the explanation, we first introduce a proposition.

**Proposition 1.** *(Convolution operations exhibit the distributive law over addition) Let* $\mathbf{x} \in \mathbb{R}^{H \times W \times C}$ *be the input feature, and let* $w_i,\ i = 1, 2, \cdots, n$ *represent the convolution kernels. The law is mathematically expressed as:*

$$\mathbf{x} * (\sum_{i=1}^{n} w_i) = \sum_{i=1}^{n} (\mathbf{x} * w_i) \tag{6}$$

*where '*' denotes the standard 2-dimensional convolution.*

The following formula then becomes evident:

$$\mathbf{x} * \mathbf{w} = \mathbf{x} * \left( \sum_{i=1}^{N} \mathbf{V_s}^i \mathbf{W}_i \right) = \sum_{i=1}^{N} \left( \mathbf{x} * (\mathbf{V_s}^i \mathbf{W}_i) \right) = \sum_{i=1}^{N} \mathbf{V_s}^i (\mathbf{x} * \mathbf{W}_i), \tag{7}$$

where $\sum_i^N \mathbf{V_s}^i = 1$ holds due to the property of Softmax operation. Essentially, the convolution with the generated weights is equivalent to a weighted sum of convolutions between the input $\mathbf{x}$ and each kernel $\mathbf{W}_i$ in the Weight Box. This process can be seen as a process of **selecting convolution kernels**. If we view each $\mathbf{W}_i$ as an expert, it's also like the Mixture-of-Experts pattern used in (Zamfir et al., 2024). For instance, with a Weight Box $\mathbf{W} \in \mathbb{R}^{2 \times P}$ that includes a low-pass $\mathbf{W}_1$ and a high-pass $\mathbf{W}_2$, the rainy input $\mathbf{x}_1$ would ideally have a larger $\mathbf{V_s}^1$ to filter out high-frequency noise, while a smaller $\mathbf{V_s}^2$ would help retain details. In contrast, for a hazy input $\mathbf{x}_2$, a larger $\mathbf{V_s}^2$ would be necessary to mitigate low-frequency haze. Although real-world degradations can be more complex, our HSN can adaptively select the appropriate weights. This insight into the selection process helps us understand HAIR, i.e., HSN produces a tailored Selecting Vector and adaptively chooses the most suitable convolution kernel for each input. Given that core operations such as convolution and matrix multiplication follow the distributive law over addition, HAIR is universal and can be integrated into various architectures such as Transformer (Vaswani et al., 2017) and Mamba (Gu & Dao, 2023).

### 3.3.2 A BRIEF ANALYSIS OF MODEL COMPLEXITY

In Section 2.2, we claim that our Hypernetworks-based method can work better than conditional embedding-based methods such as AirNet (Li et al., 2022) and PromptIR (Potlapalli et al., 2024). This section will provide a simple proof for this point. In the context of All-in-One image restoration, we aim to find a function $f : \mathbb{R}^{3HW} \to \mathbb{R}^{3HW}$ to map a degraded image $\mathbf{X}$ to a restored image $\hat{\mathbf{X}}$ by minimizing the distance between $\hat{\mathbf{X}}$ and the ground truth $\mathbf{Y} = y(\mathbf{X})$[1]. Mainstream methods typically utilize a network $g(\mathbf{X}, e(\mathbf{X}))$ to learn the mapping, where $e(\mathbf{X}) \in \mathbb{R}^k$ contains the degradation embedding of the input image, such as the prompt in PromptIR (Potlapalli et al., 2024). These methods send $e(\mathbf{X})$ together with $\mathbf{X}$ into the network to get $\hat{\mathbf{X}}$. For our HAIR, the network can be formulated as $h(\mathbf{X}; \theta(e(\mathbf{X})))$, where $\theta$ is a function that maps $e(\mathbf{X})$ into the parameters of $h$. We define the distance between the two functions as

$$d(g, y) = \min_g \max_{\mathbf{X}} \|g(\mathbf{X}, e(\mathbf{X})) - y(\mathbf{X})\|_\infty. \tag{8}$$

Given that vector functions can be complex, we define a scalar function $f^i : \mathbb{R}^M \to \mathbb{R}(i = 1, \cdots M)$ of a vector function $f : \mathbb{R}^M \to \mathbb{R}^M$. Since $f^i$ is the $i$-th element of $f$, then we have

$$\begin{aligned} d(g, y) &= \min_g \max_{\mathbf{X}} \max_i |g^i(\mathbf{X}) - y^i(\mathbf{X})| \\ &= \min_g \max_i \max_{\mathbf{X}} |g^i(\mathbf{X}) - y^i(\mathbf{X})| \\ &= \min_g \max_{\mathbf{X}} |g^{t(g)}(\mathbf{X}) - y^{t(g)}(\mathbf{X})|. \end{aligned} \tag{9}$$

Since the value range of $i$ is limited, given the function $g$, we can always find an integer $t(g)$ to replace $\max_i$. To complete the proof, we need some assumptions.

---

[1]For the sake of discussion, the following tensors are generally regarded as flattened one-dimensional vectors.

**Assumption 1.** *The target function $y^i \in \mathcal{W}_{r,3HW}, i = 1, 2, \cdots 3HW$. The Sobolev space $\mathcal{W}_{r,3HW}$ is the set of functions that are $r$-times differentiable ($r \geq 1$) with all $r$-th order derivatives being continuous and bounded by the Sobolev norm $\leq 1$, defined on $\mathbb{R}^{3HW}$.*

Intuitively, since $y^i$ maps the degraded image to a single pixel of the clean image, a slight disturbance in the degraded image should only bring a slight difference to the single pixel, so $y^i$ can be smooth and differentiable, or at least continue. Moreover, the domain and range of $y$ are restricted to $[0,1]^{3HW}$, with most pixels far less than 1, the Sobolev norm of $y_i$ can generally be less than 1 and thus this assumption generally holds.

**Assumption 2.** *For various functions (e.g. network) $g$, the integer $t(g)$ in Eq. (9) remains the same.*

In our context, $g$ represents the same network with different weights during training. No matter how the parameters update, the most "difficult" degraded image generally remains the same one, e.g., the one with very severe degradation. Within this "difficult" image, the most "difficult" pixel should also remain the same, e.g. the pixel that varies dramatically from its clean version. In this way, we simplify the vector function $g$ to scalar function $g^{t(g)}$. With the two assumptions and Theorem 2, 3, 4 and all the assumptions in (Galanti & Wolf, 2020), we can obtain the following theorems:

**Theorem 1.** *For conditional embedding-based All-in-One image restoration methods $g(\mathbf{X}, e(\mathbf{X}))$ like PromptIR (Potlapalli et al., 2024), to achieve error $d(g, y) \leq \epsilon$, the complexity (number of parameters) of the corresponding model is at least $N_g = \Omega\left(\epsilon^{-\min(3HW+k, 6HW)}\right)$, where $k$ is the dimensionality of the embedding vector. Typically $k = \Omega(HW)$ and $k \geq 3HW$.*

**Theorem 2.** *For Hypernetworks based methods $h(\mathbf{X}; \theta(e(\mathbf{X})))$ like our HAIR, to achieve error $d(h, y) \leq \epsilon$, the complexity of the corresponding model is $N_\theta = O(\epsilon^{-3HW/r})$, where $r \geq 1$.*

The two theorems demonstrate that to achieve the same error $\epsilon$, our proposed HAIR requires fewer parameters than conditional embedding-based methods as long as $\epsilon$ is small enough. Please refer to (Galanti & Wolf, 2020) and Appendix A.1 for detailed proof of Theorem 1 and 2.

## 4 EXPERIMENTS

The experimental settings follow previous research on general image restoration (Zhang et al., 2023; Potlapalli et al., 2024) in two different configurations: (a) All-in-One and (b) Single task. In the All-in-One configuration, a singular model is trained to handle multiple types of degradation, encompassing both three and five unique degradations. In contrast, the Single-task configuration involves training individual models dedicated to specific restoration tasks.

### 4.1 EXPERIMENTING PREPARATION

**Implementation Details.** Our Res-HAIR method is designed to be end-to-end trainable, eliminating the need for pre-training any of its components. The architecture comprises a 4-level encoder-decoder structure, with each level equipped with a varying number of Transformer blocks. Specifically, the number of blocks increases progressively from level-1 to level-4, following the sequence [4, 6, 6, 8]. HyperTrans Blocks are employed throughout level-4 and the decoding stages of levels 1-3. Additionally, the Weight Box parameter $N$ is set according to the sequence [5, 7, 7, 9] for each respective level. In the All-in-One setting, the model is trained with a batch size of 32, while in the single-task setting, a batch size of 8 is used. The network optimization is guided by an $L_1$ loss function, employing the AdamW optimizer (Loshchilov et al., 2017) with parameters $\beta_1 = 0.9$ and $\beta_2 = 0.999$. The learning rate is set to $2e - 4$ for the initial 150 epochs and then changed to $2e - 5$ for the final 10 epochs. To enhance the training data, input patches of size $128 \times 128$ are utilized, with random horizontal and vertical flips applied to the images to augment the dataset.

**Datasets.** We follow previous approaches (Li et al., 2022; Potlapalli et al., 2024; Zhang et al., 2023) for our All-in-One and single-task experiments, using these benchmark datasets. For single-task image denoising, we use the BSD400 (Arbelaez et al., 2010) and WED (Ma et al., 2016) datasets, adding Gaussian noise with levels $\sigma \in \{15, 25, 50\}$ to generate training images. Testing is performed on the BSD68 (Martin et al., 2001) and Urban100 (Huang et al., 2015) datasets. For deraining, we employ the Rain100L dataset (Yang et al., 2020). Dehazing experiments utilize the SOTS dataset (Li et al., 2018). Deblurring and low-light enhancement tasks use the GoPro (Nah et al., 2017) and

LOL-v1 (Wei et al., 2018) datasets, respectively. To create a comprehensive model for all tasks, we combine these datasets and train them in a setting that covers three or five types of degradations. For single-task models, training is conducted on the respective dataset.

Table 1: *Comparison to state-of-the-art on three degradations.* PSNR (dB, ↑) and SSIM (↑) metrics are reported on the full RGB images with $(^*)$ denoting methods that are not blind (need prior information of the degradation type). **Best** and second best performances are highlighted.

| Method | Params. | Dehazing | | Deraining | | Denoising | | | | | | Average | |
|---|---|---|---|---|---|---|---|---|---|---|---|---|---|
| | | SOTS | | Rain100L | | BSD68$_{\sigma=15}$ | | BSD68$_{\sigma=25}$ | | BSD68$_{\sigma=50}$ | | | |
| BRDNet (Tian et al., 2020) | - | 23.23 | .895 | 27.42 | .895 | 32.26 | .898 | 29.76 | .836 | 26.34 | .693 | 27.80 | .843 |
| LPNet (Gao et al., 2019) | - | 20.84 | .828 | 24.88 | .784 | 26.47 | .778 | 24.77 | .748 | 21.26 | .552 | 23.64 | .738 |
| FDGAN (Dong et al., 2020) | - | 24.71 | .929 | 29.89 | .933 | 30.25 | .910 | 28.81 | .868 | 26.43 | .776 | 28.02 | .883 |
| DL (Fan et al., 2019) | 2M | 26.92 | .931 | 32.62 | .931 | 33.05 | .914 | 30.41 | .861 | 26.90 | .740 | 29.98 | .876 |
| MPRNet (Zamir et al., 2021) | 16M | 25.28 | .955 | 33.57 | .954 | 33.54 | .927 | 30.89 | .880 | 27.56 | .779 | 30.17 | .899 |
| Restormer (Zamir et al., 2022) | 26M | 29.92 | .970 | 35.56 | .969 | 33.86 | .933 | 31.20 | .888 | 27.90 | .794 | 31.68 | .910 |
| AirNet (Li et al., 2022) | 9M | 27.94 | .962 | 34.90 | .967 | 33.92 | .933 | 31.26 | .888 | 28.00 | .797 | 31.20 | .910 |
| PromptIR (Potlapalli et al., 2024) | 36M | 30.58 | .974 | 36.37 | .972 | 33.98 | .933 | 31.31 | .888 | 28.06 | .799 | 32.06 | .913 |
| InstructIR$^*$ (Conde et al., 2024) | 16M | 30.22 | .959 | 37.98 | .978 | 34.15 | .933 | 31.52 | .890 | 28.30 | .804 | 32.43 | .913 |
| DaAIR (Zamfir et al., 2024) | 6M | 32.30 | .981 | 37.10 | .978 | 33.92 | .930 | 31.26 | .884 | 28.00 | .792 | 32.51 | .913 |
| Res-HAIR (*ours*) | 29M | 30.98 | .979 | 38.59 | .983 | 34.16 | .935 | 31.51 | .892 | 28.24 | .803 | 32.70 | .919 |

## 4.2 RESULTS

**All-in-One: Three degradations.** We conducted a comparative evaluation of our All-in-One image restoration model against several state-of-the-art specialized methods, including BRDNet (Tian et al., 2020), LPNet (Gao et al., 2019), FDGAN (Dong et al., 2020), DL (Fan et al., 2019), MPRNet (Zamir et al., 2021), AirNet (Li et al., 2022), PromptIR (Potlapalli et al., 2024), InstructIR (Conde et al., 2024) and DaAIR (Zamfir et al., 2024). Each of these methods was trained to handle the tasks of dehazing, deraining, and denoising. As illustrated in Table 1, our approach can significantly outperform other competing methods, e.g. an average improvement of $0.64$ dB over PromptIR. Our method notably outperforms existing benchmarks on SOTS and Rain100L datasets, by exceeding the performance of the previous best methods by margins of $0.4$ dB and $0.61$ dB respectively.

**All-in-One: Five Degradations.** Following recent studies (Zhang et al., 2023; Conde et al., 2024), we have extended our approach to deblurring and low-light image enhancement, thus extending the three-degradation setting to a more complex five-degradation setting. As demonstrated in Table 2, our method excels by learning specialized models for each degradation type while effectively capturing the shared characteristics between tasks. It achieves an average improvement of $2.03$ dB over PromptIR and $0.82$ dB over the non-blind method InstructIR, establishing a new benchmark for state-of-the-art performance across all five benchmarks. Notably, our method significantly outperforms our baseline (i.e. Restormer) in various tasks. This comparison highlights the strong capability of our method as a versatile plug-in-and-play module, enhancing the performance of the existing model with a small amount of integration complexity, e.g. only adding 3M parameters compared with Restormer.

**Single-Degradation Results.** To evaluate the effectiveness of our proposed method, we provide results in Table 3, which show the performance of individual instances of our method trained under a single degradation protocol. Specifically, the single-task variant dedicated to deraining consistently achieves higher performance than PromptIR and InstructIR by margins of $1.96$ dB and $1.02$ dB, respectively. When applied to image denoising, our method also demonstrates superiority over the aforementioned approaches, with an average improvement of $0.21$ dB and $0.47$ dB, respectively. These results underscore the significant performance improvements delivered by our method.

**Visual Results.** The visual results captured under three degradation scenarios are presented in Fig. 4. These results clearly demonstrate the superiority of our method in terms of visual quality. In the denoising task at $\sigma = 25$, Res-HAIR retains more fine details compared to other methods; in the deraining task, Res-HAIR effectively removes all rain streaks, whereas the compared methods contain obvious rain streaks; in the dehazing task, Res-HAIR may not closely resemble the ground-truth, yet it provides the most visually pleasing outcome and even clears the original haze present in the ground-truth. Overall, these visual observations confirm the efficacy of our approach in enhancing image quality across different degradation conditions.

Table 2: *Comparison to state-of-the-art on five degradations.* PSNR (dB, ↑) and SSIM (↑) metrics are reported on the full RGB images with (*) denoting methods that are not blind (need prior information of the degradation type). **Best** and **second best** performances are highlighted.

| Method | Params. | Dehazing SOTS | | Deraining Rain100L | | Denoising BSD68$_{\sigma=25}$ | | Deblurring GoPro | | Low-Light LOLv1 | | Average | |
|---|---|---|---|---|---|---|---|---|---|---|---|---|---|
| NAFNet (Chen et al., 2022a) | 17M | 25.23 | .939 | 35.56 | .967 | 31.02 | .883 | 26.53 | .808 | 20.49 | .809 | 27.76 | .881 |
| DGUNet (Mou et al., 2022) | 17M | 24.78 | .940 | 36.62 | .971 | 31.10 | .883 | 27.25 | .837 | 21.87 | .823 | 28.32 | .891 |
| SwinIR (Liang et al., 2021) | 1M | 21.50 | .891 | 30.78 | .923 | 30.59 | .868 | 24.52 | .773 | 17.81 | .723 | 25.04 | .835 |
| Restormer (Zamir et al., 2022) | 26M | 24.09 | .927 | 34.81 | .962 | 31.49 | .884 | 27.22 | .829 | 20.41 | .806 | 27.60 | .881 |
| DL (Fan et al., 2019) | 2M | 20.54 | .826 | 21.96 | .762 | 23.09 | .745 | 19.86 | .672 | 19.83 | .712 | 21.05 | .743 |
| Transweather (Valanarasu et al., 2022) | 38M | 21.32 | .885 | 29.43 | .905 | 29.00 | .841 | 25.12 | .757 | 21.21 | .792 | 25.22 | .836 |
| TAPE (Liu et al., 2022) | 1M | 22.16 | .861 | 29.67 | .904 | 30.18 | .855 | 24.47 | .763 | 18.97 | .621 | 25.09 | .801 |
| AirNet (Li et al., 2022) | 9M | 21.04 | .884 | 32.98 | .951 | 30.91 | .882 | 24.35 | .781 | 18.18 | .735 | 25.49 | .847 |
| IDR (Zhang et al., 2023) | 15M | 25.24 | .943 | 35.63 | .965 | **31.60** | .887 | 27.87 | .846 | 21.34 | .826 | 28.34 | .893 |
| InstructIR* (Conde et al., 2024) | 16M | 27.00 | .951 | **36.80** | .973 | 31.39 | **.888** | **29.73** | **.890** | 22.83 | **.836** | 30.24 | .908 |
| DaAIR (Zamfir et al., 2024) | 6M | **31.97** | **.980** | 36.28 | **.975** | 31.07 | .878 | **29.51** | **.890** | 22.38 | .825 | **30.24** | **.910** |
| Res-HAIR (*ours*) | 29M | **30.62** | **.978** | **38.11** | **.981** | 31.49 | **.891** | 28.52 | .874 | **23.12** | **.847** | **30.37** | **.914** |

Table 3: *Comparison to state-of-the-art for single degradations.* PSNR (dB, ↑) and SSIM (↑) metrics are reported on the full RGB images. **Best** and **second best** performances are highlighted.

(a) *Dehazing*

| Method | SOTS |
|---|---|
| DehazeNet (Cai et al., 2016) | 22.46 .851 |
| MSCNN (Ren et al., 2016) | 22.06 .908 |
| EPDN (Qu et al., 2019) | 22.57 .863 |
| FDGAN (Dong et al., 2020) | 23.15 .921 |
| Restormer (Zamir et al., 2022) | 30.87 .969 |
| AirNet (Li et al., 2022) | 23.18 .900 |
| PromptIR (Potlapalli et al., 2024) | 31.31 .973 |
| InstructIR (Conde et al., 2024) | 30.22 .959 |
| DaAIR Zamfir et al. (2024) | **31.99 .981** |
| Res-HAIR (*ours*) | **31.68 .980** |

(b) *Deraining*

| Method | Rain100L |
|---|---|
| DIDMDN (Zhang & Patel, 2018) | 23.79 .773 |
| UMR (Yasarla & Patel, 2019) | 32.39 .921 |
| SIRR (Wei et al., 2019) | 32.37 .926 |
| MSPFN (Jiang et al., 2020) | 33.50 .948 |
| Restormer (Zamir et al., 2022) | 36.74 .978 |
| AirNet | 34.90 .977 |
| PromptIR | 37.04 .979 |
| InstructIR | **37.98 .978** |
| DaAIR | 37.78 **.982** |
| Res-HAIR (*ours*) | **39.00 .985** |

(c) *Denoising*

| Method | σ=15 | σ=25 | σ=50 |
|---|---|---|---|
| CBM3D (Dabov et al., 2007) | 33.93 .941 | 31.36 .909 | 27.93 .833 |
| DnCNN (Zhang et al., 2017a) | 32.98 .931 | 30.81 .902 | 27.59 .833 |
| IRCNN (Zhang et al., 2017b) | 27.59 .833 | 31.20 .909 | 27.70 .840 |
| FFDNet (Zhang et al., 2018a) | 33.83 .942 | 31.40 .912 | 28.05 .848 |
| BRDNet (Tian et al., 2020) | 34.42 .946 | 31.99 .919 | 28.56 .858 |
| AirNet | 34.40 .949 | 32.10 .924 | 28.88 .871 |
| PromptIR | **34.77 .952** | **32.49 .929** | **29.39 .881** |
| InstructIR | 34.12 .945 | 31.80 .917 | 28.63 .861 |
| DaAIR | 34.55 .949 | 32.24 .924 | 29.09 .872 |
| Res-HAIR (*ours*) | **34.93 .953** | **32.70 .931** | **29.65 .885** |

## 4.3 ABLATION STUDIES

We have done several ablation studies to evaluate the effectiveness of our proposed HAIR.

**Effectiveness of Classifier and Hyper Selecting Net.** As detailed in Table 4, we study the impact of the proposed modules in four settings: *(a)* The model aligns with the Restormer architecture (Zamir et al., 2022). *(b)* In this configuration, the Global Information Vector (GIV) is directly used as conditional embeddings for the Transformer Blocks in the Decoder, rather than for weight generation. *(c)* The GIVs are designated as independently trainable parameters, each randomly initialized, with the Decoder's Transformer Blocks having their distinct GIVs. *(d)* This setup incorporates both components. The results demonstrate the indispensability of both components. With the addition of only 3M parameters and no change to the logical structure, Res-HAIR outperforms Restormer by 1.7 dB in PSNR, demonstrating its simplicity and effectiveness.

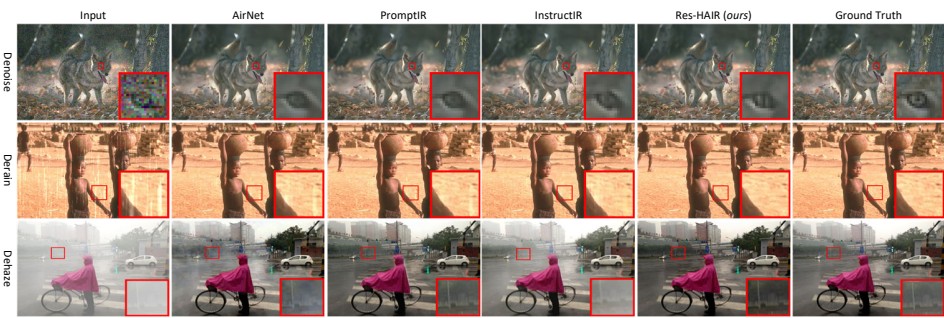

Figure 4: Visual comparison on various degradation settings.

**Position of Classifier (and HyperTrans Blocks).** In our approach, the Classifier is positioned after the first three Encoder levels, with the subsequent four Decoder levels employing HyperTrans Blocks, constituting a 3+4 configuration. The rationale for this design choice is demonstrated in Table 5. As can be seen, the optimal placement for the Classifier is at the network's midpoint. This positioning takes advantage of the expansive receptive field of the features post the network's halfway point, while allowing the HSN to dynamically generate parameters for the remaining modules. Furthermore, it is crucial to strike a balance between the receptive field size of the features fed into the Classifier and the count of HyperTrans Blocks utilized, ensuring the model's efficiency and adaptability.

Table 4: *Impact of key components.* Results are from single deraining task on Rain100L.

| Setting | Prams. | *Classifier* | *HSN* | PSNR | SSIM |
|---|---|---|---|---|---|
| *(a)(baseline)* | 26M | ✗ | ✗ | 36.74 | .978 |
| *(b)* | 27M | ✓ | ✗ | 36.88 | .979 |
| *(c)* | 28M | ✗ | ✓ | 36.76 | .979 |
| *(d) (ours)* | 29M | ✓ | ✓ | **39.00** | **.985** |

Table 5: *Impact of position.* Results are from Denoise task on Urban100 ($\sigma$=25).

| Setting | (1+6) | (2+5) | (4+3) | (5+2) | (6+1) | (3+4) *(ours)* |
|---|---|---|---|---|---|---|
| PSNR | 32.45 | 32.56 | 32.68 | 32.61 | 32.51 | **32.70** |
| SSIM | 0.927 | 0.929 | 0.931 | 0.930 | 0.927 | **0.931** |

**HAIR for Different Baselines.** We have previously posited that HAIR is essentially a plug-in-and-play module which is readily integrable with any existing network architecture. To substantiate this claim, we have implemented HAIR on various baselines. As depicted in Table 6, we selected three efficacious image restoration models, i.e. Transweather (Valanarasu et al., 2022), AirNet (Li et al., 2022), and Restormer (Zamir et al., 2022) for integration with HAIR. Specifically, we have integrated the Classifier at the network's midpoint for each method and transitioned the subsequent layers to Hypernetworks-based modules. The results show that our HAIR can significantly improve the performance of these baselines. Additionally, our HAIR outperforms PromptIR as a plug-in-and-play module, demonstrating its better application value.

Table 6: *HAIR for different baseline architectures.* PSNR (dB, ↑) and SSIM (↑) metrics are reported on the full RGB images. **Best** performances are highlighted.

| Method | Main Operation | Params | *Dehazing* SOTS | | *Deraining* Rain100L | | *Denoising* BSD68$_{\sigma=25}$ | | *Deblurring* GoPro | | *Low-Light* LOLv1 | | Average | |
|---|---|---|---|---|---|---|---|---|---|---|---|---|---|---|
| Transweather | | 38M | 21.32 | .885 | 29.43 | .905 | 29.00 | .841 | 25.12 | .757 | 21.21 | .792 | 25.22 | .836 |
| Transweather+PromptIR | Self-Attention | 51M | 22.89 | .920 | 29.79 | .913 | 29.95 | .877 | 25.74 | .781 | 23.02 | .849 | 26.28 | .868 |
| Transweather+HAIR | | 42M | **23.66** | **.935** | **32.34** | **.947** | **29.96** | **.875** | **26.33** | **.802** | **23.16** | **.858** | **27.09** | **.884** |
| AirNet | | 9M | 21.04 | .884 | 32.98 | .951 | 30.91 | .882 | 24.35 | .781 | 18.18 | .735 | 25.49 | .847 |
| AirNet+PromptIR | Convolution | 14M | 21.34 | .883 | 33.52 | .953 | 30.92 | .882 | 24.37 | .786 | 18.18 | .737 | 25.67 | .848 |
| AirNet+HAIR | | 10M | **22.15** | **.899** | **34.56** | **.957** | **30.94** | **.884** | **25.44** | **.792** | **18.24** | **.740** | **26.27** | **.854** |
| Restormer | | 26M | 24.09 | .927 | 34.81 | .962 | 31.49 | .884 | 27.22 | .829 | 20.41 | .806 | 27.60 | .881 |
| Restormer+PromptIR | Convolution | 36M | 30.61 | .974 | 36.17 | .973 | 31.25 | .887 | 27.93 | .851 | 22.89 | .842 | 29.77 | .905 |
| Restormer+HAIR | | 29M | **30.62** | **.978** | **38.11** | **.981** | **31.49** | **.891** | **28.52** | **.874** | **23.12** | **.847** | **30.37** | **.914** |

## 5    CONCLUSION & FUTURE PROSPECT

This paper introduces HAIR, a novel plug-and-play Hypernetworks-based module capable of being easily integrated and adaptively generating parameters for different networks based on the input image. Our method comprises two main components: the Classifier and the Hyper Selecting Net (HSN). Specifically, the Classifier is a simple image classification network with Global Average Pooling, designed to produce a Global Information Vector (GIV) that encapsulates the global information from the input image. The HSN functions as a fully-connected neural network, receiving the GIV and outputting parameters for the corresponding modules. Extensive experiments indicate that HAIR can significantly enhance the performance of various image restoration architectures at a low cost without necessitating any changes to their logical structures. By incorporating HAIR into the widely recognized Restormer architecture, we have achieved State-Of-The-Art performance on a range of image restoration tasks. The potential for further exploiting data-conditioned Hypernetworks in tasks such as image restoration, editing, and generation is substantial, given their robust adaptability to diverse inputs over the mainstream conditional embedding techniques.

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

# A APPENDIX

## A.1 DETAILED PROOF OF THEOREM 1, THEOREM 2

**Notations** We consider $\mathcal{X} = [-1, 1]^{m_1}$ and $\mathcal{I} = [-1, 1]^{m_2}$ and denote, $m := m_1 + m_2$. Here $\mathcal{X}$ stands for the set of input images $\mathbf{X}$, meanwhile $\mathcal{I}$ refers to the set of degradation information $e(\mathbf{X})$. For a closed set $X \subset \mathbb{R}^n$, we denote by $C^r(X)$ the linear space of all $r$-continuously differentiable functions $h : X \to \mathbb{R}$ on $X$ equipped with the supremum norm $\|h\|_\infty := \max_{x \in X} \|h(x)\|_1$.

**Theorem 3.** *(The **Theorem 2** in (Galanti & Wolf, 2020)) Let $\sigma$ be a universal, piece-wise $C^1(\mathbb{R})$ activation function with $\sigma' \in BV(\mathbb{R})$ and $\sigma(0) = 0$. Let $\mathcal{E}_{\mathbf{e},\mathbf{q}}$ be a neural embedding method. Assume that $\mathbf{e}$ is a class of continuously differentiable neural network $e$ with zero biases, output dimension $k = \mathcal{O}(1)$ and $\mathcal{C}(e) \le \ell_1$ and $\mathbf{q}$ is a class of neural networks $q$ with $\sigma$ activations and $\mathcal{C}(q) \le \ell_2$. Let $\mathbb{Y} := \mathcal{W}_{1,m}$. Assume that any non-constant $y \in \mathbb{Y}$ cannot be represented as a neural network with $\sigma$ activations. If the embedding method achieves error $d(\mathcal{E}_{\mathbf{e},\mathbf{q}}, \mathbb{Y}) \le \epsilon$, then, the complexity of $\mathbf{q}$ is:*
$N_{\mathbf{q}} = \Omega\left(\epsilon^{-(m_1+m_2)}\right)$.

The notation $BV(\mathbb{R})$ stands for the set of functions of bounded variation,

$$BV(\mathbb{R}) := \left\{ f \in L^1(\mathbb{R}) \mid \|f\|_{BV} < \infty \right\} \text{ where, } \|f\|_{BV} := \sup_{\substack{\phi \in C_c^1(\mathbb{R}) \\ \|\phi\|_\infty \le 1}} \int_{\mathbb{R}} f(x) \cdot \phi(x) \, \mathrm{d}x \qquad (10)$$

Note that a distinct neural network $e$ is not mandatory. For example, the "prompts" in PromptIR (Potlapalli et al., 2024) are a set of trainable parameters that do not require a separate network to generate them. Yet, the conclusion remains the same even if network $e$ is non-existent.

**Theorem 4.** *(The **Theorem 3** in (Galanti & Wolf, 2020)) In the setting of Theorem 3, except $k$ is not necessarily $\mathcal{O}(1)$. Assume that the first layer of any $q \in \mathbf{q}$ is bounded $\|W^1\|_1 \le c$, for some constant $c > 0$. If the embedding method achieves error $d(\mathcal{E}_{\mathbf{e},\mathbf{q}}, \mathbb{Y}) \le \epsilon$, then, the complexity of $\mathbf{q}$ is:*
$N_{\mathbf{q}} = \Omega\left(\epsilon^{-\min(m, 2m_1)}\right)$.

**Theorem 5.** *(The **Theorem 4** in (Galanti & Wolf, 2020)) Let $\sigma$ be as in Theorem 3. Let $y \in \mathbb{Y} = \mathcal{W}_{r,m}$ be a function, such that, $y_I$ cannot be represented as a neural network with $\sigma$ activations for all $I \in \mathcal{I}$. Then, there is a class, $\mathbf{g}$, of neural networks with $\sigma$ activations and a network $f(I; \theta_f)$ with ReLU activations, such that, $h(x, I) = g(x; f(I; \theta_f))$ achieves error $\le \epsilon$ in approximating $y$ and $N_{\mathbf{g}} = \mathcal{O}\left(\epsilon^{-m_1/r}\right)$.*

In the realm of image restoration, $m_1$ equals $3HW$, and $m_2$ equals $k$, where $k$ denotes the dimensionality of the flattened degradation embedding. In our method, $k$ is consistent with the shape of the Global Information Vector (GIV), specifically $2C$, and thus is $\mathcal{O}(1)$. Conversely, in PromptIR (Potlapalli et al., 2024), $k$ is dynamic and contingent on the resolution of the input image, precluding it from being $\mathcal{O}(1)$. Theorem 5 indicates that the complexity for a Hypernetworks-based method to attain an error of $\epsilon$ is $\mathcal{O}\left(\epsilon^{-3HW/r}\right)$. Theorems 3 and 4 collectively suggest that the complexity for embedding-based methods is at least $\Omega\left(\epsilon^{-\min(3HW+k, 6HW)}\right)$. This comparison illustrates that Hypernetworks-based methods like HAIR may require fewer parameters to reach a given error threshold compared to their embedding-based counterparts.

## A.2 DISCUSSION

### A.2.1 HAIR FOR CONFLICTING DEGRADATIONS.

As previously discussed in the Introduction (Section 1), the performance of conventional All-in-One image restoration methods, which rely on a single model with static parameters, can be significantly compromised when dealing with conflicting image degradations. To demonstrate this, we trained models on various combinations of datasets, each representing different types of degradations, and subsequently evaluated these models on their respective benchmarks. The outcomes are presented in Tables 7 and 8.

From a frequency domain analysis perspective, haze is identified as low-frequency noise, whereas rain and Gaussian noise are categorized as high-frequency disturbances. Conflicts arise when the

degradations in a combined scenario include both low- and high-frequency noise components. According to Table 7, it is evident that the presence of conflicting degradations, such as the combination of noise and haze or rain and haze, can severely degrade the model's performance. In contrast, when the combined degradations do not conflict, like the combination of noise and rain, the performance loss is minimal and may even enhance the overall performance.

Table 8 exhibits a similar trend, but with a notably reduced impact from conflicting degradations. This reduction in performance impairment underscores the effectiveness of our proposed Hypernetworks-based approach, which dynamically generates parameters based on the input image's content. This adaptability allows our method to mitigate the performance loss typically associated with conflicting degradations.

Table 7: Performance of the PromptIR (Potlapalli et al., 2024), when trained on different combinations of degradation types (tasks) i.e., removal of gaussian noise, rain and haze. Note that the "combination" here stands for combination of Datasets with single degradations instead of composite degradations. "Conflicting" here shows if the combined degradations are conflicting to each other. PSNR/SSIM are reported.

| Degradation | | | Conflicting | Denoising on BSD68 dataset | | | Deraining on Rain100 | Dehazing on SOTS |
|---|---|---|---|---|---|---|---|---|
| Noise | Rain | Haze | | $\sigma = 15$ | $\sigma = 25$ | $\sigma = 50$ | | |
| ✓ | ✗ | ✗ | - | 34.34/0.938 | 31.71/0.898 | 28.49/0.813 | - | - |
| ✗ | ✓ | ✗ | - | - | - | - | 37.04/0.979 | - |
| ✗ | ✗ | ✓ | - | - | - | - | - | 31.31/0.973 |
| ✓ | ✓ | ✗ | ✗ | 34.26/0.937 | 31.61/0.895 | 28.37/0.810 | 39.32/0.986 | - |
| ✓ | ✗ | ✓ | ✓ | 33.69/0.928 | 31.03/0.880 | 27.74/0.777 | - | 30.09/0.975 |
| ✗ | ✓ | ✓ | ✓ | - | - | - | 35.12/0.969 | 30.36/0.973 |
| ✓ | ✓ | ✓ | ✓ | 33.98/0.933 | 31.31/0.888 | 28.06/0.799 | 36.37/0.972 | 30.58/0.974 |

Table 8: Performance of the our proposed Res-HAIR, when trained on different combinations of degradation types (tasks) i.e., removal of gaussian noise, rain and haze. PSNR/SSIM are reported.

| Degradation | | | Conflicting | Denoising on BSD68 dataset | | | Deraining on Rain100 | Dehazing on SOTS |
|---|---|---|---|---|---|---|---|---|
| Noise | Rain | Haze | | $\sigma = 15$ | $\sigma = 25$ | $\sigma = 50$ | | |
| ✓ | ✗ | ✗ | - | 34.36/0.938 | 31.72/0.898 | 28.50/0.813 | - | - |
| ✗ | ✓ | ✗ | - | - | - | - | 39.00/0.985 | - |
| ✗ | ✗ | ✓ | - | - | - | - | - | 31.68/0.980 |
| ✓ | ✓ | ✗ | ✗ | 34.33/0.937 | 31.66/0.896 | 28.44/0.811 | 41.55/0.989 | - |
| ✓ | ✗ | ✓ | ✓ | 34.13/0.935 | 31.49/0.892 | 28.22/0.803 | - | 31.18/0.979 |
| ✗ | ✓ | ✓ | ✓ | - | - | - | 38.44/0.983 | 31.22/0.979 |
| ✓ | ✓ | ✓ | ✓ | 34.16/0.935 | 31.51/0.892 | 28.24/0.803 | 38.59/0.983 | 30.98/0.979 |

### A.2.2 HAIR FOR UNSEEN COMPOSITE DEGRADATION.

Since HAIR can accurately discriminate unseen composite degradations, as illustrated in Fig. 2d, we test Res-HAIR with these settings, as shown in Table 9, 11, 10. The results somehow shows the generalization ability of our proposed method. However, we consider the restoration outcomes for such degradations are not satisfactory enough. The interesting fact is, sometimes HAIR generates GIVs that are intermediate when confronted with composite degradations. This tendency results in weights that are a midpoint between those associated with each individual degradation type. For example, an image with both noise and haze degradations results in weights that are intermediate between the weights for images affected by noise alone and those affected by haze alone, which fails to fully eliminate either degradation and result in a somehow unsatisfactory performance. This insight reveals HAIR's operational mechanism, highlighting its strategy for weights generation.

Table 9: Results of unseen composite degradation (haze + noise) on SOTS.

| Method | PSNR | SSIM |
|---|---|---|
| PromptIR | 16.211 | 0.785 |
| Res-HAIR | **16.798** | **0.802** |

Table 10: Results of unseen composite degradation (rain + noise) on Rain100L.

| Method | PSNR | SSIM |
|---|---|---|
| PromptIR | 24.348 | 0.726 |
| Res-HAIR | **24.365** | **0.729** |

Table 11: Results of unseen composite degradation (rain + haze) on SOTS.

| Method | PSNR | SSIM |
|---|---|---|
| PromptIR | 23.784 | 0.686 |
| Res-HAIR | **23.823** | **0.692** |

## A.3 POSSIBLE CONFUSIONS

### A.3.1 DOES HAIR ALSO UTILIZE FIXED PARAMETERS AS PREVIOUS METHODS DO?

In the Introduction (Section 1), we highlighted that our proposed HAIR method differs from previous approaches, which typically employ a static set of parameters to handle various image degradations. In contrast, HAIR employs a data-conditioned Hypernetwork to dynamically generate parameters based on the input image's content. However, a potential point of confusion arises from the fact that the Hypernetworks in HAIR (i.e., the Hyper Selecting Nets) are indeed fixed during inference, suggesting that HAIR also relies on a set of fixed parameters to address different degradations.

To clarify this confusion, it is crucial to understand our motivation for using Hypernetworks. Our goal is to mitigate the performance loss caused by conflicting degradations. For example, by generating distinct parameters, we aim to enable the model to function either as a low-frequency or high-frequency filter, depending on the input image's requirements. As illustrated in Fig. 1, the primary focus should be on the main networks that directly process the image.

Therefore, when we refer to "fixed parameters" and "dynamic parameters," we are referring to the parameters of the main network that interacts with the image, not the Hypernetworks responsible for parameter generation, which do not directly engage with the input image. The point lies in the dynamic adaptation of the main network's parameters to the specific characteristics of the input image, which is the innovative aspect of HAIR.

## A.4 MORE IMPLEMENTATION DETAILS

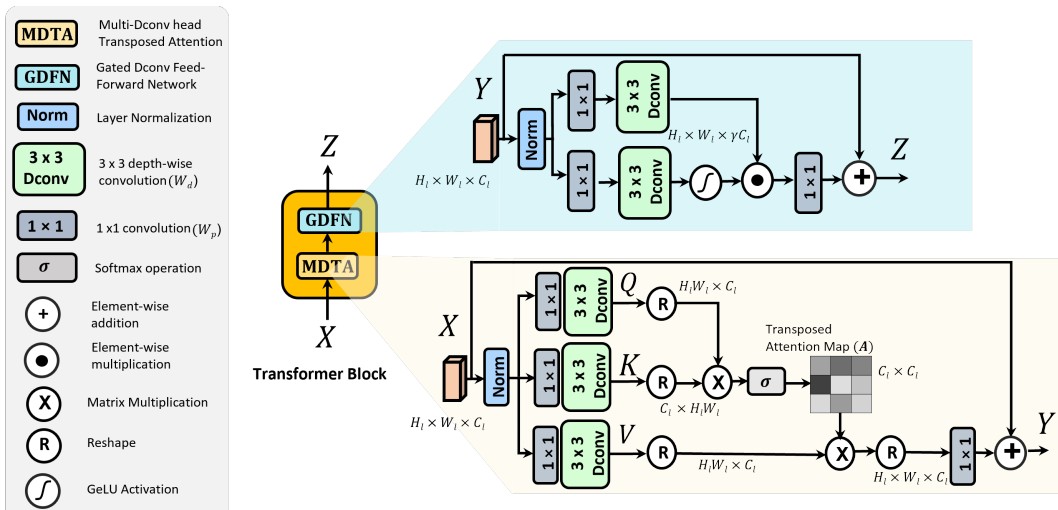

Figure 5: Overview of the Transformer block used in the Res-HAIR framework. The Transformer block is composed of two sub modules,the Multi Dconv head transposed attention module(MDTA) and the Gated Dconv feed-forward network(GDFN).

### A.4.1 Transformer Block in Res-HAIR Framework

(This section is directly borrowed from the paper of PromptIR. (Potlapalli et al., 2024)) In this section, we present the block diagram5 of the transformer block and further, elaborate on the details of the transformer block employed in the Res-HAIR framework. The transformer block follows the design and hyper-parameters outlined in (Zamir et al., 2022)

To begin, the input features $\mathbf{X} \in \mathbb{R}^{H_l \times W_l \times C_l}$ are passed through the MDTA module. In this module, the features are initially normalized using Layer normalization. Subsequently, a combination of $1 \times 1$ convolutions followed by $3 \times 3$ depth-wise convolutions are applied to project the features into Query ($\mathbf{Q}$), Key ($\mathbf{K}$), and Value ($\mathbf{V}$) tensors. An essential characteristic of the MDTA module is its computation of attention across the channel dimensions, rather than the spatial dimensions. This effectively reduces the computational overhead. To achieve this channel-wise attention calculation, the $Q$ and $K$ projections are reshaped from $H_l \times W_l \times C_l$ to $H_l W_l \times C_l$ and $C_l \times H_l W_l$ respectively, before calculating dot-product, hence the resulting transposed attention map with the shape of $C_l \times C_l$. Bias-free convolutions are utilized within this submodule. Furthermore, attention is computed in a multi-head manner in parallel.

After MDTA Module the features are processed through the GDFN module. In the GDFN module, the input features are expanded by a factor $\gamma$ using $1 \times 1$ convolution and they are then passed through $3 \times 3$ convolutions. These operations are performed through two parallel paths and the output of one of the paths is activated using GeLU non-linearity. This activated feature map is then combined with the output of the other path using element-wise product.

### A.4.2 Implementation Details of HyperTrans Blocks

This section elaborates on specific details presented in Section 3.1.2. Within the Restormer framework (Zamir et al., 2022), each Transformer consists of a total of 12 logical convolutional layers, as depicted in Fig. 5. For code implementation, we utilize 6 convolutional layers to effectively emulate the functionality of 12 convolutional layers within each Transformer Block. Consequently, the generated parameters $\mathbf{w} \in \mathbb{R}^P$ described in Section 3.1.2 encompass all the parameters necessary for these 6 convolutional layers of a single Transformer Block. The Weights Box $\mathbf{W} \in \mathbb{R}^{N \times P}$, therefore, represents the aggregate parameters for $N$ Transformer Blocks.

Given a fixed Global Information Vector (GIV) $\mathbf{V_g}$, it is processed through the Fully-Connected Neural Network (FCNN) followed by a Softmax operation to yield the Selecting Vector $\mathbf{V_s} \in \mathbb{R}^N$. This vector is subsequently employed to "select" the definitive parameters from the Weights Box. Each Decoder level possesses a distinct Weights Box, which is shared among all the HyperTrans Blocks at that level. It is crucial to highlight that all HyperTrans Blocks across all levels operate using the same GIV derived from a single Classifier. Meanwhile, each single HyperTrans Block is equipped with its own Hyper Selecting Net, resulting in a unique Selecting Vector and, consequently, its own set of parameters for each.

### A.5 Visual Results

In Fig. 6 and Fig. 7 we provide more visual results to show the strong ability of our method for image restoration tasks.

972
973
974
975
976
977
978
979
980
981
982
983
984
985
986
987
988
989
990
991
992
993
994
995
996
997
998
999
1000
1001
1002
1003
1004
1005
1006
1007
1008
1009
1010
1011
1012
1013
1014
1015
1016
1017
1018
1019
1020
1021
1022
1023
1024
1025

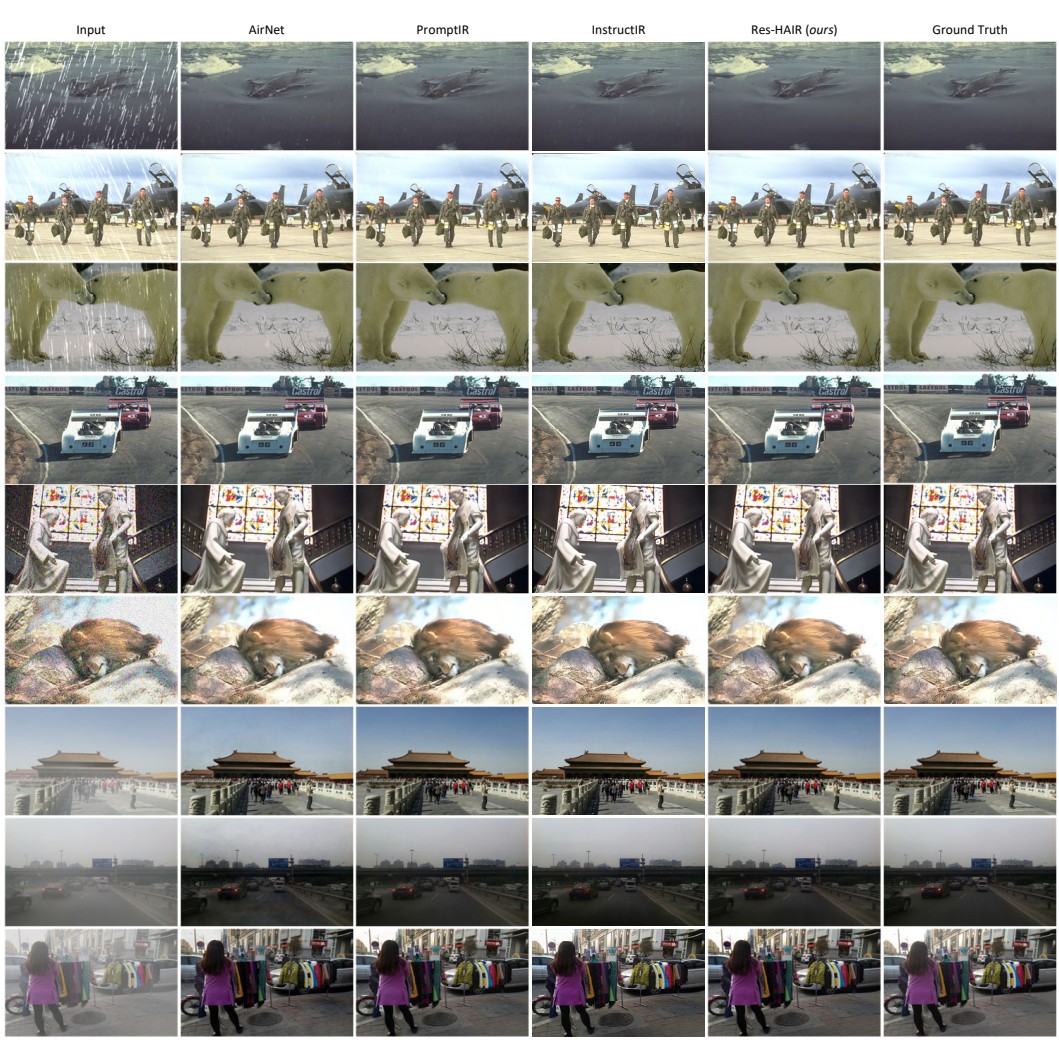

Figure 6: Additional visual results.

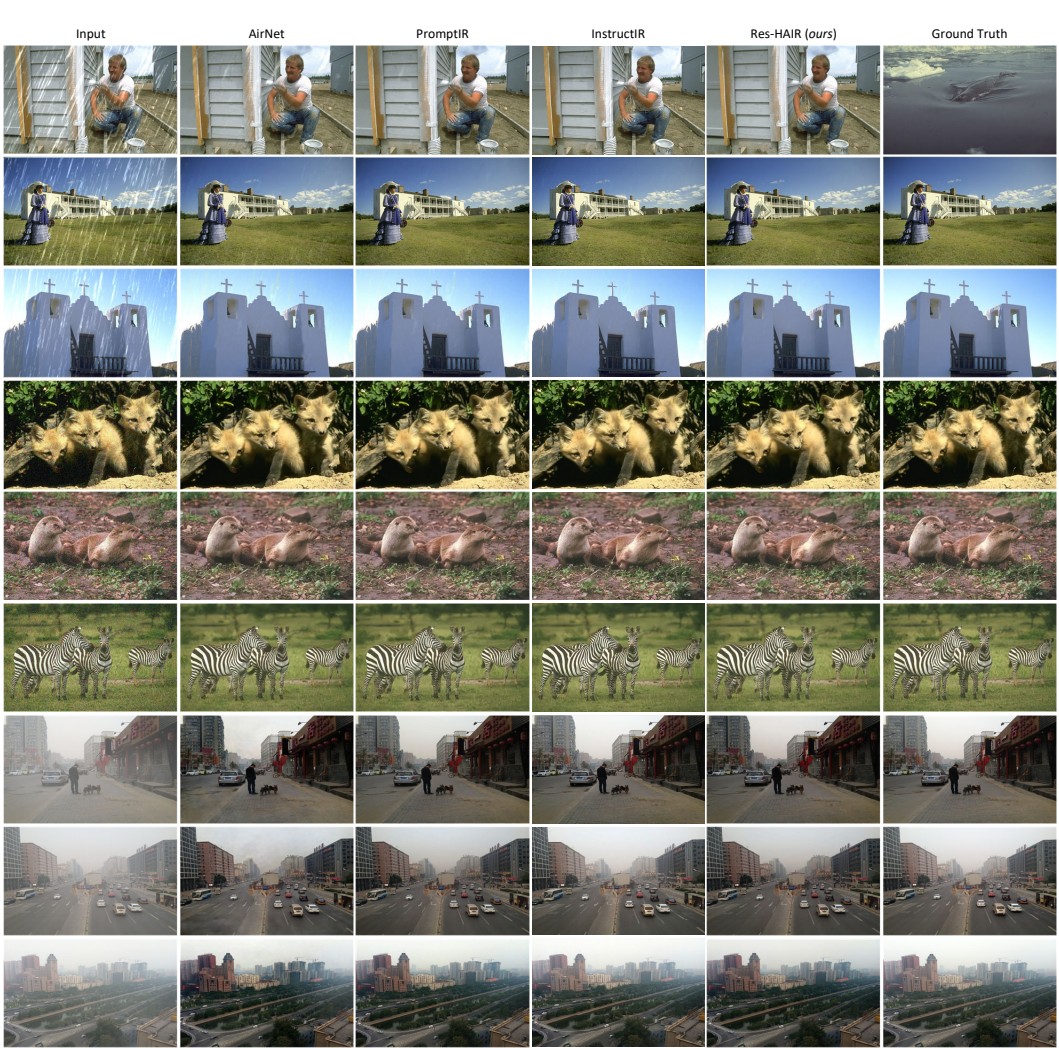

Figure 7: Additional visual results

