# OpenReview forum: "HAIR: Hypernetworks-based All-in-One Image Restoration"
_ICLR.cc/2025/Conference — ICLR 2025 Conference Withdrawn Submission_

### Official Review · Reviewer_X8Rw · 2024-10-29

**Soundness:** 3
**Presentation:** 2
**Contribution:** 2
**Rating:** 3
**Confidence:** 5

**Summary:**

This paper proposes a hypernetworks-based all-in-one image restoration plug-and-play method called HAIR. This method generates parameters based on the input image, allowing the model to dynamically adapt to specific degradations. HAIR consists of two main components，the Classifier and the Hyper Selecting Net. The Classifier is a simple image classification network that generates a Global Information Vector containing the degradation information of the input image. The HSN is a fully connected neural network that receives the GIV and outputs parameters for the corresponding modules.

**Strengths:**

1. The paper introduces HAIR, a novel approach that leverages hypernetworks to dynamically generate parameters for image restoration tasks. This represents an innovative application of hypernetworks in the field of image restoration, offering a fresh perspective. The method is designed to be plug-and-play, allowing integration with existing image restoration models to enhance their performance without significant structural changes. The authors have also provided code for supplementary materials, which is a positive aspect for reproducibility.
2. The paper is well-organized and structured, guiding the reader through the problem, the proposed solutions, experiments, and results. Each section transitions logically to the next, making the paper easy to follow.

**Weaknesses:**

1. While the paper claims that HAIR can generalize to unseen composite degradations, the results for these cases show only marginal improvements over the baseline PromptIR method. The restoration outcomes for such degradations are not satisfactory, indicating that the model may not be fully capturing the complexities of real-world degradation combinations.

2. The paper emphasizes the parameter efficiency of HAIR compared to embedding-based methods. However, there is a lack of discussion on the computational cost and runtime of the hypernetwork-generated parameters. In practical applications, especially in resource-constrained environments, both the number of parameters and the computational efficiency are crucial.

3. The experiments are comprehensive but are limited to a specific set of benchmark datasets. To further validate the generalization capabilities of HAIR, it would be beneficial to include additional datasets that capture a wider variety of real-world conditions and degradation types.  Furthermore, the paper does not provide a more comprehensive visual comparison.

4. The ablation studies are limited to the impact of the Classifier and HSN components. More detailed ablation studies on other aspects, such as the depth of the Hyper Selecting Net or the dimensionality of the Global Information Vector, could provide deeper insights into the factors that influence HAIR's performance.

**Questions:**

1. Could the authors elaborate on how the classifier module performs classification? For example, is it based on degradation types such as denoising or deraining, or does it classify based on internal features or another method? A detailed explanation would be helpful. Additionally, are the authors familiar with the PromptIR model? The results in Table 3 indicate that there is little difference between Res-HAIR and PromptIR. Have the authors validated the effectiveness and computational efficiency of these two networks?

2. The ablation study is limited to the impact of the classifier and HSN components. It is suggested that the authors investigate the influence of the depth of the Hyper Selecting Network or the dimension of the Global Information Vector on the model's performance. Additionally, in Table 4, the Rain100L dataset is used for the ablation study, while the de-raining results in Table 2 also employ the Rain100L dataset. Could the authors explain the discrepancy in the obtained PSNR results?

3. Could the authors discuss potential practical applications of HAIR, especially in scenarios where images are degraded by multiple factors, such as in autonomous driving or surveillance systems? How does HAIR's performance translate to these real-world applications?

---

### Official Review · Reviewer_e5U5 · 2024-11-01

**Soundness:** 3
**Presentation:** 3
**Contribution:** 3
**Rating:** 6
**Confidence:** 4

**Summary:**

The paper proposes HAIR, a Hypernetworks-based all-in-one image restoration method that dynamically generates parameters based on input images. Unlike existing methods that use fixed parameters for different degradations, HAIR adapts to each specific degradation, enhancing its effectiveness. By incorporating a Classifier and Hyper Selecting Net, HAIR outperforms current state-of-the-art models like PromptIR and InstructIR, achieving superior or comparable results.

**Strengths:**

1. Unlike traditional approaches that use a fixed set of parameters for handling different types of degradation, HAIR dynamically generates parameters tailored to the specific input image. This adaptive parameter generation is an innovative use of hypernetworks in image restoration, marking a significant departure from the existing fixed-parameter paradigm.
2. The manuscript is written in a clear and concise manner, making it easy to follow. The logical flow of the content aids in understanding the key points.
3. HAIR's significance lies in its potential to redefine how All-in-One image restoration tasks are approached. By moving away from the conventional fixed-parameter models to an adaptive, data-conditioned mechanism, HAIR not only enhances restoration quality but also simplifies model design, making it more suitable for real-world applications where degradations are often unknown and mixed.

**Weaknesses:**

1. While the Hyper Selecting Net effectively generates dynamic parameters, it does introduce additional computational complexity, especially during inference. A more in-depth analysis of this trade-off between performance improvement and computational cost could further strengthen the claims.

**Questions:**

1. Could you provide more detailed insights into the computational overhead introduced by the Hyper Selecting Net during inference? Specifically, how does the increased parameter generation cost compare with conventional fixed-parameter models, particularly regarding runtime and GPU memory usage? This would help in better understanding the practical deployment challenges of HAIR.

---

### Official Review · Reviewer_us7z · 2024-11-03

**Soundness:** 3
**Presentation:** 3
**Contribution:** 2
**Rating:** 5
**Confidence:** 5

**Summary:**

This work proposes a plug-and-play Hypernetworks-based module, which can be easily integrated and adaptively generate parameters for differentnet works based on the input image. Specifically, two main components are introduced, the Classifier and the Hyper Selecting Net (HSN) . The former is a simple image classification network with Global Average Pooling, designed to produce a Global Information Vector (GIV) that encapsulates the global information from the input image. The HSN functions as a fully-connected neural network, receiving the GIV and outputting parameters for the corresponding modules.

**Strengths:**

1.This work theoretically prove that, for a given small enough error threshold ϵ in image restoration tasks, HAIR requires fewer parameters compared to main stream embedding-based All-in-One methods.

2. The writing is logically rigorous.

**Weaknesses:**

1. Some experiments are not provided, such as the single-task performance usually tested in all-in-one image restoration works as an performance upper bound, and the results in real-world cases in addition to the synthetic cases.

2. The analysis should be improved, such as the dehazing performance of the proposed method over DaAIR.

3. The descriptions for the proposed method should be improved, such as the loss function.

More details of the weakness are provided in the Questions part.

**Questions:**

1. The proposed method may not appear dominant for the dehaze task when compared to DaAIR. The authors should consider analyzing the reasons for this.

2. More visual results are needed for tasks such as deblurring and low-light enhancement. Additionally, quantitative comparisons for single-degradation experiments (deblurring and low-light enhancement) should be included. The current visual results make it challenging to observe differences, so the authors could consider labeling these differences in the plots or displaying PSNR/SSIM metrics to improve clarity.

3. An updated baseline could be added to Table 6 to strengthen comparisons.

4. What does "Linear Combination" refer to in Figure 3? How is the "weights box" generated?

5. The article lacks a description of the loss function. Additionally, is there a loss constraint applied to the Global Information Vector (GIV)?

6. Consider adding single-task performance as an upper bound in Tables 9, 10, and 11 to verify the authors' conjecture: “This tendency results in weights that are a midpoint between those associated with each individual degradation type.”

7. Since the experiments in this paper are based on synthetic data, the authors could consider validating the model on real images to demonstrate its generalization ability.

8. Some comparison methods in Tables 3 and 6 lack citations.

---

### Note · Authors · 2024-11-13

**Comment:**

We sincerely appreciate the detailed and constructive feedback provided by all three reviewers. However, given that low-level vision is a relatively niche area within the ICLR community, we feel that some reviewers may not be fully familiar with its specific technical aspects. After careful consideration, we have decided to withdraw our paper.

**Withdrawal Confirmation:**

I have read and agree with the venue's withdrawal policy on behalf of myself and my co-authors.